# Study on the Interference Law of AC Transmission Lines on the Cathodic Protection Potential of Long-Distance Transmission Pipelines

**Boyang Zhang \*, Lin Li, Yansong Zhang and Jielin Wang**

Key Laboratory of Shaanxi Province for Gas-Oil Logging Technology, Xi'an Shiyou University,
Xi'an 710065, China
\* Correspondence: 20211030292@stumail.xsyu.edu.cn

**Abstract:** Through inductive coupling, AC transmission lines can generate large amounts of voltage to buried oil and gas pipelines in areas with common corridors, posing a threat to the cathodic protection effect of pipelines. Therefore, this paper investigates the effect of AC transmission lines on the cathodic protection of long-distance pipelines through inductive coupling. COMSOL Multiphysics finite element simulation software is used to calculate the distribution of cathodic protection potential of long-distance pipelines under different voltage levels, parallel spacing, conductor-to-ground height, conductor arrangement and pipeline burial depth for normal operation of AC transmission lines. Comparison and analysis of the AC transmission line on the pipeline cathodic protection potential interference law is conducted. The results show that: 1. AC transmission lines cause serious electromagnetic interference with the pipeline cathodic protection system, which will cause the pipeline cathodic protection potential to shift out of the effective protection area. 2. The maximum value of the induced voltage of the pipeline will appear at the two ends of the pipeline, and the induced voltage of the pipeline in the middle position is 0. 3. The shift of the pipeline cathodic protection potential increases with the increase of voltage level and decreases with the increase of parallel spacing, conductor height and burial depth. The pipeline cathodic protection potential shift is highest when the wires are arranged horizontally and lowest when they are arranged in an umbrella shape.

**Keywords:** AC transmission line; inductive coupling; long-distance pipeline; cathodic protection potential; COMSOL

## 1. Introduction

Long-distance pipeline transportation is the main mode of transportation for gas and oil. With the rapid development of modern industry, the length of the pipeline is increasing, but the routing restrictions, the phenomenon of parallel and cross between the pipeline and transmission lines is difficult to avoid. At this time, transmission lines can be formed in the pipeline through resistive coupling, inductive coupling and capacitive coupling, such as induced voltage and stray current [1]. When serious, the interference will alter the cathodic protection effect on the pipeline and accelerate the pipeline's corrosion [2].

When AC transmission lines cross or are parallel with the pipeline, the transmission lines can interfere with the pipeline cathodic protection potential through capacitive coupling, resistive coupling and magnetic induction coupling. The resistive coupling is mainly due to the short-circuit current flowing into the soil through the grounding pole to form stray currents on the cathodic protection potential of the pipeline. Capacitive coupling refers to the existence of alternating electric fields around high-voltage transmission conductors when alternating currents are flowing in them and are caused by electrostatic conditions. The role of induction in the adjacent pipeline makes it easy to couple out the induced voltage. In general, oil and gas pipelines are buried in deep soil. The capacitance

of the soil is small and due to the shielding effect of the earth, capacitive coupling generally does not act on the pipeline [3,4]. When the transmission line is in normal operation, magnetic induction coupling is the main reason for interference of the AC transmission line with the cathodic protection potential. Magnetic induction coupling refers to the alternating electromagnetic field generated by the AC transmission line. The pipeline cuts the magnetic induction line, resulting in the inductive electric potential on the pipeline (Figure 1), which causes the cathodic protection potential to shift and affects the cathodic protection effect on the long-distance transmission pipeline [5].

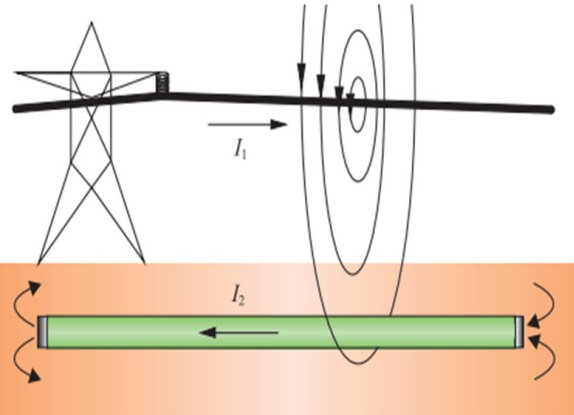

**Figure 1.** Induced effects of pipelines and AC transmission lines.

Only the case of magnetic induction coupling is considered here, i.e., the induced voltage on the pipeline from the steady operating current during normal operation of the transmission line. Excessive induced voltages can cause the reference potential of a pipeline with applied current protection methods to shift out of the protected zone and can accelerate the corrosion of the pipeline.

Scholars from various countries have used different methods to conduct in-depth studies on the electromagnetic coupling of transmission lines to buried pipelines. To calculate the electromagnetic coupling effect of transmission lines on adjacent buried oil and gas pipelines, it is usually necessary to calculate the earth return impedance of a complex conductor system consisting of lines and pipelines. Carson and Pollaczek proposed the earth return impedance formula between conductors under the homogeneous hook soil model in 1926 [6], and in 1969, Sunde derived the earth return impedance formula for overhead conductor systems under the two-layer soil model based on Carson's formula [7]. In 2015, Ametani et al. further investigated how Carson's formula handles displacement currents based on layered earth impedance using the earth return conductor, and the results showed that Carson's formula can handle displacement currents when the relative earth dielectric constant is 1 [8]. Taflove et al. in [9] used transmission line theory to predict the voltage induced on a gas transmission pipeline by a 60 Hz AC transmission line sharing a joint right-of-way. Davinin equivalent circuits for the pipeline section were developed, allowing the decomposition of complex pipeline power line geometries. Al-Alawi et al. [10] proposed an artificial neural network (ANN) model-based method for predicting the electromagnetic inference effects of shared rights-of-way (ROW) for gas pipelines of high-voltage transmission lines. In [11], Xuan et al. proposed an analytical method for calculating transient induced voltages and currents due to inductive coupling on a pipe using a circuit model. The waveforms of transient induced voltages and currents on the pipe were verified by numerical arithmetic examples. In [12], Levente Czumbil et al. proposed the use of an artificial intelligence (AI) approach applied to the study of electromagnetic interference problems between high-voltage overhead power lines (HV OPL) and nearby underground metallic pipes (MP). An artificial neural network (ANN) solution was implemented to evaluate the inductive coupling matrix that described the OPL-MP electromagnetic interference problem in the presence of different problem

geometries and multi-layer soil structures.Wang Chenyang et al. proposed a method to determine the minimum horizontal spacing between transmission lines and underground pipelines for reducing induced electromagnetic interference under normal operation of transmission lines. A soil model is created for each location using the RESAP module in the CDEGS software package and once the soil model is determined, the induced AC voltage is simulated using the HIFREQ module in CDEGS and a minimum spacing distance can be suggested. Four voltage levels of the transmission line were considered and two curves and their empirical models were created for each voltage level to determine the minimum separation distances [13].

Most of the current research on the electromagnetic interference of AC transmission lines to buried pipelines does not take into account the existence of cathodic protection systems, while in actual engineering, in order to have a stable and safe operation of pipelines, long-distance oil and gas pipelines must be protected by cathodic protection systems. Therefore, it is necessary to study the impact of AC transmission lines on the cathodic protection potential of long-distance pipelines. Most of the current research on this problem also uses the CDEGS simulation software based on the method of moments (MOM) theory, while the COMSOL Multiphysics simulation software based on the finite element method is rarely used in the study of this problem. Therefore, this paper uses COMSOL Multiphysics simulation software to establish an accurate simulation model and analyze the interference law of AC transmission lines on the cathodic protection potential of pipelines based on the finite element method (FEM). This can be used as a basis for the design of buried lines and cathodic protection schemes for long-distance transmission pipelines.

In this paper, the interference law of AC transmission line on pipeline cathodic protection potential is analyzed based on the finite element method (FEM) using COMSOL Multiphysics simulation software. The cathodic protection system model based on the applied current method is established using the secondary current distribution interface in COMSOL Multiphysics and solved using a steady-state solver to obtain the pipeline potential distribution in the absence of electromagnetic interference. Then, the magnetic field and current (Def) interfaces in the AC/DC module are used to model the electromagnetic interference. Finally, the above model is solved using the frequency domain solver. The cathodic protection potential distribution of the pipeline after interference was obtained.

## 2. Numerical Calculation Methods and Comparison of Common Electromagnetic Fields

There are many numerical methods for calculating electromagnetic fields, and the solution methods vary for different problems, but Maxwell's equations are the basis for the numerical calculation of electromagnetic fields. With the increasing speed of computers, numerical methods have been widely used and developed rapidly, and a large number of complex electromagnetic field problems have been solved by numerical calculation methods. The common ones are simulated charge method, boundary element method, method of moments, finite element method, etc.

The analog charge method is based on the uniqueness theorem of the electric field, whereby a continuous free charge on the surface of an electrode or a continuous bound charge on the dividing surface of a dielectric is replaced by a set of discrete analog charges of equal value. In this way, the discrete analog charges are superimposed in space by applying the superposition principle to obtain the spatial electric field distribution generated by the original continuously distributed charges.

The boundary element method is a numerical method for solving differential equations based on classical integral equations and the finite element method. The basic idea is that the corresponding fundamental solution of the differential equation is used as the weight function. The weighted residual method is applied, followed by the Green's function, to derive the integral equation that relates the value of the function to be solved to the value of the function and the value of the normal guide on the boundary. The integral equation is made to hold the boundary to obtain the boundary integral element. The boundary element method uses the boundary integral equation as the starting point to solve for the unknown

quantity on the boundary. On the basis of the derived boundary integral equation, the boundary is discretized by using the idea of discretization of finite elements; after solving, the function values and normal guide values of all the nodes on the boundary can be obtained. By substituting all the boundary values into the integral equation, the expression of the function values of the inner points can be obtained, which can be expressed as a linear combination of the values of the boundary nodes. The main disadvantage of the boundary element method is that the coefficient matrix of the system of boundary element equations is an asymmetric full array, and the method is currently applicable only to linear problems.

The method of moments is based on the weighted residual method for solving differential equations, and integral equations that are applicable to a method. In general, the method of moments has several calculation steps. First, according to the original function of the problem to be solved, find its approximate function solution, which can be substituted into the operator equation to obtain a set of residual functions. When the residual function and a set of test functions have been selected for the orthogonal, you can find a matrix that can be solved to obtain the solution of the function. During this process, the selection of the basis function and check function is a key factor in the solution, and selecting the appropriate functions will directly affect both the workload and the accuracy of the calculation.

The finite element method is a numerical calculation method based on the integral expression method, and it operates on the principle of variational differentiation and partial interpolation. It transforms the required differential equation-type mathematical model-side-value problem, into a corresponding variational problem, i.e., the general function to find the extreme value. However, instead of solving the partial differential equation directly, it dissects the field into several small and simple cells by means of grid dissection and cell interpolation before selecting the basis functions. Difference cells have different coefficients of the basis functions, thus discretizing the variational problem from a continuous coal mass into a multifunctional polar-value problem with a finite number of variables. The basis functions and the functions to be solved from them are then only required to be continuous within the cell, i.e., have piecewise continuity. The functions to be solved must satisfy the continuity condition on simple boundaries between cells rather than needing to form a full domain basis function. This allows an approximate solution to this mathematical physics problem to be generated by solving a system of algebraic equations.

In terms of computational time, the moment and boundary element methods are relatively long because they require the solution of a full array matrix. The finite element method and the simulated charge method are relatively short in terms of computation time because they only require solving sparse matrices.

Considered in terms of hardware requirements, the hardware requirements for the method of moments depend mainly on the number of basis functions and the size of the linear system of equations. In general, the method of moments requires a large amount of memory and storage space as well as a high computational speed. The hardware requirements for the finite element method depend mainly on the number and type of cells and the accuracy of the approximate solution. In general, the finite element method requires more processor cores and parallel computing power as well as a higher floating-point speed. The hardware requirements for the boundary element method depend mainly on the number of boundary cells or nodes and the size of the set of integral equations or matrix equations. In general, the boundary element method requires comparable or slightly less memory and storage space than the method of moments, and comparable or slightly higher computational speed than the finite element method.

In terms of accuracy of results, the finite element method and the simulated charge method have relatively high accuracy of results as they can be adapted to complex geometries and nonlinear problems. The results of the momentum and boundary element methods are relatively less accurate as they are only suitable for simple geometries and linear problems.

In summary, the finite element method has the following advantages. It is more flexible in meshing and has good accuracy. It can construct many types of cells, and when meshing a field, one type of cell can be used or many types of cells can be combined; each type of cell has a different shape, so no matter how complex the boundary surfaces and intersections of the field are, the adaptive meshing method can be applied. The coefficient matrices of the FEM equations are sparsely symmetric, which makes the solution process easier, less time consuming and less memory intensive. The boundary conditions are easily incorporated into the mathematical model of the FEM, reducing the difficulty of writing computer programs. The finite element method also has outstanding advantages in dealing with complex problems, such as complex geometries of field boundaries and boundary conditions with different media partitioning interfaces. These advantages have caused the finite element method to become the leading numerical calculation method for dealing with various complex electromagnetic field problems.

## 3. Applied Current Method

The method of protecting the pipe by applying a current can also be referred to as the forced current method (Figure 2), which uses an external DC source as the source of supply for polarization. Usually, an FeSi anode is connected to the positive end of the power supply, and the negative end of the power supply is connected to the target pipe body. This causes current to flow through the auxiliary anode to the pipe and back to the power supply from the cathodic contact point of the pipe, causing cathodic polarization of the pipe and achieving the protective effect [14].

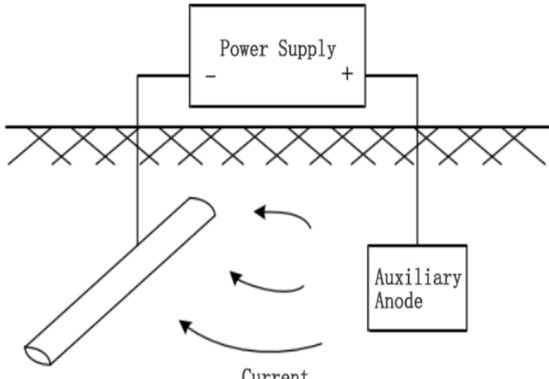

**Figure 2.** Schematic diagram of the external current protection method.

The system has several main working principles. To adjust the output voltage and output current according to the given potential and the reference electrode potential, maintain the pipe ground potential of the protected metal body relative to the copper sulfate reference electrode in the range of −850~−1200 mV [15], and ensure that the cathodic polarization potential between the metal surface of the protected body of the pipe and the reference electrode in contact with the soil is not less than 100 mV, so that the protected metal is continuously polarized under the cathodic action. Ensure that the cathodic polarization potential between the metal surface of the protected pipeline and the reference electrode in contact with the soil is not less than 100 mV so that the protected metal is continuously polarized under the action of the cathodic current. This significantly reduces the corrosion rate of the protected pipeline and achieves the purpose of cathodic protection.

The external current protection method is applicable to the cathodic protection of long-distance pipelines. The method can control the size of the pipeline protection current by adjusting the output of the external power supply to keep the pipeline voltage within a reasonable protection zone.

## 4. Theoretical Foundations of Numerical Calculation Methods for Work-Frequency Electromagnetic Fields

### 4.1. Differential Equations for Electromagnetic Fields

Maxwell's system of equations is a set of partial differential equations that describe the relationship between the electric and magnetic fields and the density of charge and current. Maxwell's system of equations is the basis for the study and analysis of electromagnetic fields. Maxwell's system of equations is made up of four laws: Ampere's law of loops, Faraday's law of electromagnetic induction, Gauss' law of electric flux and Gauss' law of magnetic flux [16].

The differential forms of Ampere's law of loops, Faraday's law of electromagnetic induction, Gauss's law of electric flux and Gauss's law of magnetic flux, are expressed as follows:

$$\nabla \times H = J + \frac{\partial D}{\partial t} \tag{1}$$

$$\nabla \times E = \frac{\partial B}{\partial t} \tag{2}$$

$$\nabla D = \rho \tag{3}$$

$$\nabla B = 0 \tag{4}$$

where $H$ is the magnetic field strength (A/m), $J$ is the current density (A/m$^2$), $D$ is the electric flux density (C/m$^2$), $E$ is the electric field strength (V/m), $B$ is the magnetic induction strength (T) and $\rho$ is the charge body density (C/m$^3$).

For the calculation of electromagnetic fields, in order to simplify the problem, we usually introduce two auxiliary quantities to separate the electric and magnetic field variables, forming a separate partial differential equation for the electric or magnetic field respectively, which facilitates the numerical solution. These two auxiliary quantities are the vector magnetic potential $A$ and the scalar potential $\varphi$, which are defined as:

Vector magnetic potential

$$B = \nabla \times A \tag{5}$$

Scalar potential

$$E = -\nabla \varphi \tag{6}$$

The vector magnetic potential A and the scalar potential φ are collectively known as the electromagnetic dynamic potential.

In accordance with the above, the vector magnetic potential $A$ and the scalar potential $\varphi$ can automatically satisfy Faraday's law of electromagnetic induction and Gauss's law of magnetic flux. Applying them to Ampere's law of loops and Gauss' law of flux, the partial differential equations for the magnetic field and the partial differential equations for the electric field, respectively, can be derived as follows:

$$\nabla^2 A - \mu\varepsilon\frac{\partial^2 A}{\partial t^2} = -\mu J \tag{7}$$

$$\nabla^2 \varphi - \mu\varepsilon\frac{\partial^2 \varphi}{\partial t^2} = -\frac{\rho}{\varepsilon} \tag{8}$$

where $\mu$ and $\varepsilon$ are the magnetic permeability and dielectric constant of the medium, respectively, and $\nabla^2$ are the Laplace operators [17,18]:

$$\nabla^2 = \left(\frac{\partial^2}{\partial x^2} + \frac{\partial^2}{\partial y^2} + \frac{\partial^2}{\partial z^2}\right) \tag{9}$$

It is easy to see from this that Equations (7) and (8) have the same form and are symmetric to each other, which suggests that the same approach can be used in solving the magnetic field partial differential equation and the electric field partial differential equation.

Since the relationship between $A$ and $\varphi$ satisfies the Laurenz condition:

$$\nabla \bullet A = -\mu\varepsilon\frac{\partial\varphi}{\partial t} \tag{10}$$

when solving the finite element method numerically, the potential function equation is used as the basic equation for the calculation, and after the unknowns, $A$ and $\varphi$ are obtained, the values of the field distribution of the electric or magnetic field intensity are then solved from their defining equations. Then, after various transformations (i.e., post-processing by the software), the passive electrostatic field and the constant magnetic field potential function can be reduced to Laplace's equation, the active electrostatic field can be reduced to Poisson's equation, and the time-harmonic problem can be reduced to Helmholtz's equation, which can be solved to obtain the various physical quantities in the electromagnetic field of interest.

*4.2. Boundary Conditions for Electromagnetic Field Analysis*

In the practical solution of electromagnetic field problems, the equations have a definite solution only if they are subject to definite boundary conditions and initial conditions. There are three main types of boundary conditions for electromagnetic field problems: the Dirichlet boundary condition, the Neumann boundary condition and a combination of these boundary conditions.

(1)    Dirichlet boundary conditions

The Dirichlet boundary condition can be expressed as follows:

$$\varphi|_\Gamma = g(\Gamma) \tag{11}$$

where $\Gamma$ is the Dirichlet boundary, and $g(\Gamma)$ is the position function and can be constant or zero. When $g(\Gamma)$ is zero, the Dirichlet boundary is said to be an odd boundary condition, such as assuming that the potential of one pole of a parallel plate capacitor is zero and the other is constant; the boundary condition where the potential is zero is an odd boundary condition.

(2)    Neumann (Neumann) boundary conditions

The Neumann (Neumann) boundary condition can be expressed as

$$\frac{\delta\varphi}{\delta n}\bigg|_\Gamma + f(\Gamma)\varphi\bigg|_\Gamma = h(\Gamma) \tag{12}$$

where $\Gamma$ is the Newman boundary, $n$ is the outer normal vector of the boundary $\Gamma$, $f(\Gamma)$ and $h(\Gamma)$ are general functions whose values can be taken as constant or zero, and when taken as zero, this Newman boundary condition is the odd Newman boundary condition.

## 5. Simulation Model Building and Calculation PARAMETERS

COMSOL Multiphysics is a numerical simulation software based on the finite element method [19], which enables the simulation of real physical phenomena by solving partial differential equations (single field) or partial differential equations (multiple fields) on the basis of the finite element method.

*5.1. Geometric Modeling*

The modeling of the geometric model is done using COMSOL (Figure 3), which consists of a pipe, transmission conductor, auxiliary anode, soil, and air domain. The top half of the model is the three transmission lines and the air domain, the bottom half is the pipe, and the auxiliary anode buried in the soil domain. Add a layer 0.1 m above the bottom of the soil domain and define the very small domain formed by this layer and the bottom of the soil as the infinite source domain, which can be regarded as infinite and has

a uniform soil resistivity. Set the bottom surface of the soil domain as the potential zero, i.e., the potential zero is defined at the infinity of the soil. The size of the air domain is 400 m × 40 m × 100 m, the size of the auxiliary anode is 25 m × 1 m × 5 m, and a total of 10 groups are buried. The selected pipe length is 400 m and the transmission line stall distance is 400 m.

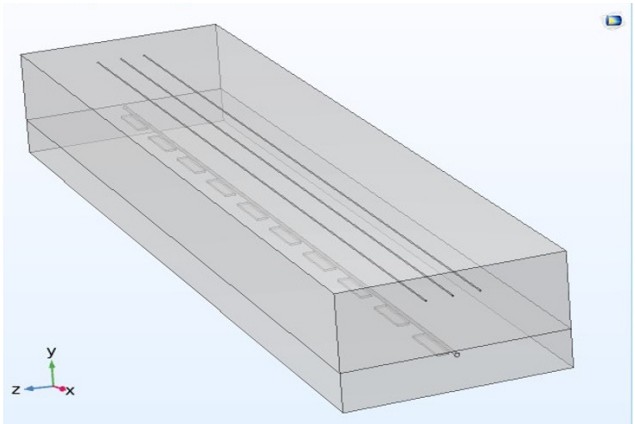

**Figure 3.** Geometric model diagram.

### 5.2. Grid Division

The model is meshed using a global adaptive mesh refinement method, which determines the point with the largest local error in the modeling domain by means of an error estimation strategy. Then, the FEM analysis software generates a completely new mesh based on the error estimation information. The software takes into account the local errors of the entire model while using smaller cells in the regions with relatively large local errors. The complete mesh contains "127,484" domain cells, "13,987" boundary cells and "5625" edge cells (Figure 4).

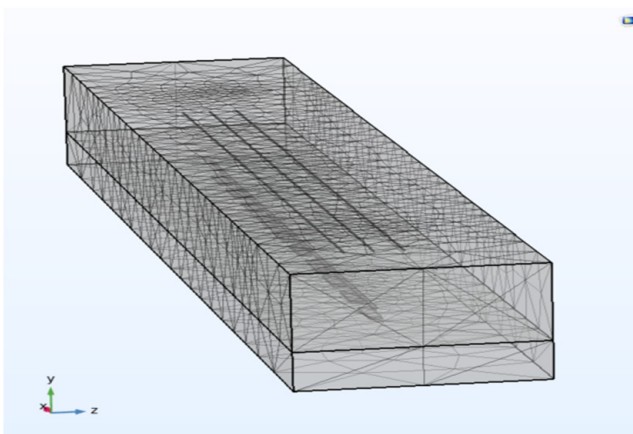

**Figure 4.** Grid division diagram.

### 5.3. Relevant Calculation Parameters

The height of the conductor was taken as 18 m, 28 m and 38 m; the voltage level was taken as 500 kV, 750 kV and 1000 kV; the parallel spacing was taken as 0 m, 15 m and 25 m; the buried depth was taken as 2 m, 4 m and 6 m, the conductor arrangement was taken as horizontal, umbrella and triangle [20]. Soil conductivity of 0.2 S/m in the parallel section of the line and pipe.

### 5.3.1. Transmission Line Conductors

The transmission conductor is LGJ-400/50 with an outside diameter of 27.63 mm and an electrical conductivity of $3.54 \times 10^7$ S/m and a magnetic conductivity of $\mu_0$.

### 5.3.2. Operating Current

Generally speaking, the higher the voltage level of the transmission line, the higher the operating current. The common operating currents (Table 1) in transmission lines of 500 kV, 750 kV and 1000 kV are selected for the calculation and studied respectively, and the frequency was 50 Hz [21]. Three phase conductors are placed from left to right and labeled phase A, phase B and phase C. The phases of the three conductors are arranged in positive order, with each phase differing by 120° in turn, i.e., phase A 0°, phase B −120° and phase C 120°.

**Table 1.** Operating currents of different levels of transmission lines.

| Transmission Line Voltage Levels | Operating Current |
| --- | --- |
| 500 kV | 2000 A |
| 750 kV | 3000 A |
| 1000 kV | 4000 A |

Pipeline Parameters: Tables 2 and 3.

**Table 2.** Pipeline parameters.

| Length | Pipe Diameter | Materials | Wall Thickness | Relative Resistivity | Relative Magnetic Permeability |
| --- | --- | --- | --- | --- | --- |
| 400 m | 1067 mm | Steel | 14.27 mm | 10 | 636 |

**Table 3.** Actual measured AC-induced voltage values versus simulated calculated AC-induced voltage values.

| Test Pile Number | AC-Induced Voltage (Measurement) | AC Interference Voltage (Simulation) |
| --- | --- | --- |
| 1 | 24.39 V | 22.64 V |
| 2 | 31.87 V | 32.21 V |
| 3 | 44.47 V | 44.89 V |
| 4 | 13.82 V | 14.59 V |
| 5 | 6.49 V | 5.81 V |
| 6 | 16.65 V | 19.02 V |
| 7 | 5.35 V | 4.31 V |
| 8 | 11.89 V | 12.77 V |
| 9 | 24.86 V | 24.14 V |
| 10 | 30.87 V | 30.51 V |

### 5.4. Validation of the Model

Figure 5 shows the distribution of a section of the transmission line and the long transmission pipeline in the 1000 kV Southeast Jinan–Nanyang–Jingmen EHV AC test demonstration project in China. The red lines indicate the buried route of the pipeline, the green line segments indicate the transmission line erection route, and the blue dots indicate the test pile locations. The authors went to the site to measure parameters related to electromagnetic induction in long-distance transmission pipelines, such as the height of transmission lines, soil resistivity and burial depth of pipelines, and collected the induction voltages of pipelines measured at different cathodic protection test stakes. The collected parameters were substituted into the established COMSOL model for simulation calculations and compared with the data measured at the test piles.

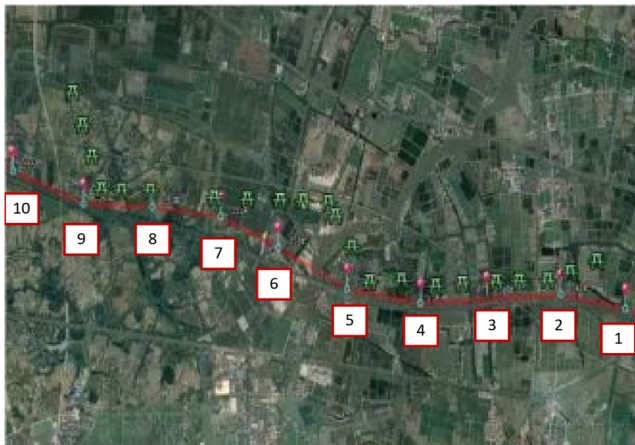

**Figure 5.** Distribution of power transmission lines and long-distance pipelines.

The test stakes are numbered 1 to 10 in sequence from right to left. A peak AC-induced voltage of 44.47 V was present on the first pipe at test pile 3, and a peak AC-induced voltage of 16.65 V was present on the second pipe under test pile 6; a low value of 6.49 V was measured on the first AC induced voltage at test pile 5, and a low value of 5.35 V was measured on the second pipe AC induced voltage at test pile 7. The simulated results were compared with the measured voltages in the field, and the results showed that the two predicted peak AC induction voltages were 44.89 V and 19.02 V, respectively, which differed from the measured data in the field by 0.42 V and 2.37 V. The two simulated low AC induction voltages were 4.81 V and 3.31 V, which differed from the measured data in the field by 1.68 V and 2.04 V. The error was within the permissible range, and the simulation could be used to continue the study. The model can be used to continue the study.

## 6. Analysis of the Electromagnetic Influence of AC Transmission Lines on the Cathodic Protection Potential of Long-Distance Transmission Pipelines

The interference pattern of the AC transmission line on the cathodic protection potential of the pipeline was analyzed based on the finite element method (FEM) using COMSOL Multiphysics simulation software. A model of the cathodic protection system based on the applied current method was developed using the secondary current distribution interface in COMSOL Multiphysics and solved using a steady-state solver to obtain a map of the pipeline potential distribution in the absence of electromagnetic interference. Then, the magnetic field and current (Def) interfaces in the AC/DC module were used to model the electromagnetic disturbance. Finally, the above model was solved using a frequency domain solver. The cathodic protection potential distribution of the pipeline after the interference was obtained.

### 6.1. Simulation of Pipeline Cathodic Protection

The secondary current distribution interface in the COMSOL Multiphysics electrochemical corrosion module was used to build a cathodic protection system using the applied current protection method. Calculated parameters, such as equilibrium potential, exchange current density, Tafel slope and oxygen reference concentration, are brought into the kinetics that are built-in to the Tafel electrode kinetic equations for cathodes and Butler–Volmer polarization. The kinetic principle boundary conditions for the steady state calculation of the electrode reaction at the interface between the metal material and the electrolyte are also involved. The pipeline potential distribution when the pipeline is not subjected to AC electromagnetic interference can be obtained, and the pipeline potential is within the effective protection zone of −0.85 V to −1.2 V (Figure 6).

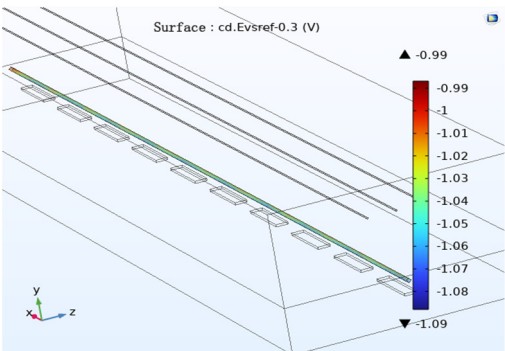

**Figure 6.** Distribution of cathodic protection potential of the pipeline.

### 6.1.1. Control Engineering Equations for Cathodic Protection Systems

The problem of cathodic protection of pipelines is primarily a problem of the electric field of the pipeline in an electrostatic field (i.e., steady state field). In order to design a cathodic protection system, the expected current density or electrochemical potential on the metal surface needs to be known. Numerical simulation methods can therefore be carried out to assist and evaluate alternative design situations [22].

The choice of boundary conditions in the solution region determines the complexity of solving the control equations.

Figure 7 shows the schematic diagram of the boundary element of the buried pipe, where the whole electrolyte domain (soil) is surrounded by the boundary.

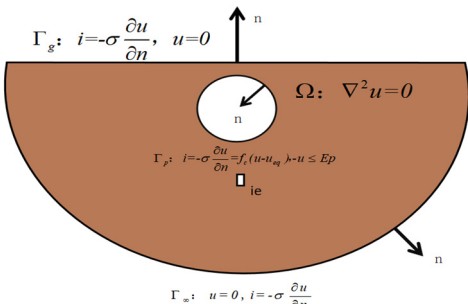

**Figure 7.** Schematic diagram of buried pipeline boundary.

Where $\Omega$ is the electrolyte (soil) domain; $\Gamma_g$ is the normal derivative of the ground and surface boundary; $\Gamma_p$ is the outer boundary of the buried metal pipe; $U_{eq}$ is the natural corrosion potential of the pipe in the soil; $\Gamma_\infty$ is the infinity of the soil; $\Gamma_a$ is the silicon-iron anode boundary; and f $(u - u_{eq})$ is the cathodic polarization function.

Since the length of the pipe selected for this study is long at 400 m, the resistance and the current path of the pipe as an electrode material cannot be neglected, and the potential distribution $\Phi_{met}$ within the electrode material obeys the Laplace equation.

$$\nabla \cdot (k_{met} \nabla \Phi_{met}) = 0 \tag{13}$$

where $k_{met}$ is the conductivity of the electrode material to its connected circuit. The thermodynamic driving force for the electrochemical reaction at the metal-soil interface is written as Equation (14).

$$V = \Phi_{met} - \Phi_{sol} \tag{14}$$

The two structural domains, the electrode material and the electrolyte, are connected by charge conversion through electrode dynamics, which yields:

$$k_{met} n \cdot \nabla \Phi_{met} = k_{sol} n \cdot \nabla \Phi_{sol} \tag{15}$$

where $k_{sol}$ is the conductivity of the electrolyte, expressed as:

$$k_{sol} = F^2 z_i u_i c_i \tag{16}$$

where $F$ is the Faraday constant, 96.485 C/eq, $Z_i$ is the charge of substance $i$, $u_i$ is the mobility of substance $i$, and $C_i$ is the concentration of substance i.

In order to solve Equations (13), all boundaries in the system require boundary conditions of either the fundamental type ($\Phi = C1$) or the natural one ($n \bullet \nabla \Phi = C_2$) (C1 and C2 may be constants or functions). For the electrodes, the model considers the bare metal as well as the anode polarization kinetics. The insulator can be considered as having zero normal gradients, i.e., $n \bullet \nabla \Phi = 0$.

In order to consider the potential drop of the anode, a simple polarization model was used for the anode's conditions, which takes into account corrosion and oxygen reduction.

$$i_a = i_{o_2}(10^{\frac{\Phi_{met} - \Phi_{in} - \Phi_{corr}}{\beta_{anode}}} - 1) \tag{17}$$

where $i_{O2}$ is the mass transfer-limited current density for oxygen reduction, $\Phi_{Corr}$ is the free corrosion potential of the anode, and $\beta$ is the Tafel slope of the anodic corrosion reaction.

Considering the computational complexity, the boundary conditions are simplified as follows.

(1) Soil interface at infinity $\Gamma_\infty$: Since the current cannot flow from the surface to the air, the surface is an insulating boundary, i.e., i = 0. At the same time, the normal guide value of the anode to the infinite soil surface, i.e., the current density, is also zero.

(2) Cathode boundary condition f(u − u_eq): that is, the boundary condition $\Gamma_p$ of the pipeline is defined by using the polarization function that describes the functional relationship between the current density i and the potential u. i = fc(u − u_eq), (or the polarization concentration curve) because the polarization curve is a simulation of the physical behavior of the electrolyte–metal interface, which is nonlinear. There are also nonlinear terms in the equation when it is being solved, so it must be treated using the Newton–Raphson iterative method and the segmented linear fitting method.

(3) Anode boundary $\Gamma_a$: The boundary assumes that it satisfies the first type of boundary condition, i.e., the current flowing out of the anode is a constant value. The current anode flows through the soil to the pipe, so the soil potential at infinity is almost zero.

The simplified control equations and boundary conditions of the cathodic protection system for oil and gas pipelines in the electrostatic field are shown in Equation (18).

$$\begin{aligned}
\Omega &: \nabla^2 u(x) = b(x) \\
\Gamma_\infty &: u(x) = 0, i = -\sigma \frac{\partial u}{\partial n} = 0 \\
\Gamma_g &: i(x) = -\sigma \frac{\partial u}{\partial n} = 0 \\
\Gamma_p &: i(x) = -\sigma \frac{\partial u}{\partial n} = f_c(u - u_{eq}) \\
\Gamma_a &: i(x) = i_a
\end{aligned} \tag{18}$$

where $u(x)$ is the potential function (defined with reference to the electrode saturated CuSO$_4$ (SCE)); $i(x)$ is the normal derivative of the potential function, i.e., the current density; $f_c$ denotes the polarization curve, $u_{eq}$ is the self-corrosion potential of the pipe in the soil medium; $\Gamma = \Gamma_\infty + \Gamma_g + \Gamma_p + \Gamma_a$ denotes the boundary of the computational domain $\Omega$, $n$ is the normal vector of the boundary $\Gamma$; and $\sigma$ is the soil conductivity, then $\nabla^2$ is the Laplace operator.

6.1.2. Establishment of Boundary Conditions

The entire soil domain was set as an electrolyte domain, the six outer surfaces of the soil domain and the outer surface of the auxiliary anode were set as insulated, and the initial potential of the electrolyte was 0. The pipe and the FeSi anode were set as electrode surface 1 and electrode surface 2 respectively. The outer potential of electrode surface 1 was

$-2$ V and the in situ density was $1 \times 10^{-5}$ mol/m$^2$. The outer potential of electrode surface 2 was 0 V and the in situ density was $1 \times 10^{-5}$ mol/m$^2$.

There are several mathematical forms available to describe the interfacial reaction process, such as the Tafel empirical formula and the Butler–Volmer equation, etc. [23]. In the present research, the Tafel empirical formula was employed. For the anodic process, the main process considered is iron oxidation, and the governing equation is as follows:

$$i_{a,Fe} = i_{o,Fe} 10^{\frac{E - E_{eq,Fe}}{A_{Fe}}} \tag{19}$$

where $i_{a,Fe}$ refers to the corrosion current density, $io_{,Fe}$ designates the exchange current density, $E$ stands for the polarization potential, $E_{eq,Fe}$ indicates the equilibrium potential and $A_{Fe}$ is the Tafel slope. For the cathodic process, the main process may contain the reduction process of oxygen and water depending on the cathodic polarization potential. Normally, oxygen reduction is dominant. Thus, based on the conventional Tafel empirical formula, we modified the oxygen concentration. The governing equation is as follows [23]:

$$i_{c,O_2} = \frac{c_{O_2}}{c_{O_2\_ref}} i_{o,O_2} 10^{\frac{E - E_{eq,O_2}}{A_{O_2}}} \tag{20}$$

where $c_{O2}$ refers to the interfacial oxygen concentration and $c_{O2\_ref}$ is the reference concentration of oxygen in pore solution of soil, which is set as 8.6 mol/m$^3$ [23]. The physical meaning of other parameters is similar to Equation (19). The reduction of the water process can be depicted in the form of Equation (19), which will not be repeated here. The corresponding parameters have been presented in Table 4 [23].

**Table 4.** Calculation parameters.

| Parameters | Values | Description |
|---|---|---|
| A_Fe | 0.41 V | Tafel slope iron oxidation |
| A_H$_2$ | $-0.15$ V | Tafel slope hydrogen evolution |
| A_O$_2$ | $-0.18$ V | Tafel slope oxygen reduction |
| C_O$_2$_ref | 8.6 mol/m$^3$ | Oxygen reference concentration |
| Eeq_Fe | $-0.76$ V | Iron oxidation equilibrium potential |
| Eeq_H$_2$ | $-1.03$ V | Hydrogen evolution equilibrium potential |
| Eeq_O$_2$ | 0.189 V | Oxygen reduction equilibrium potential |
| i0_Fe | $7.1 \times 10^{-5}$ A/m$^2$ | Iron oxidation exchange current density |
| i0_H$_2$ | $1.1 \times 10^{-3}$ A/m$^2$ | Hydrogen evolution current density |
| i0_O$_2$ | $7.7 \times 10^{-7}$ A/m$^2$ | Oxygen reduction exchange current |
| Eeq_FeSi | $-0.68$ V | FeSi equilibrium potential |

### 6.2. Effect of the Voltage Level on the Cathodic Protection Potential of Pipelines

To study the influence of voltage level on cathodic protection potential, the initial condition includes a burial depth of 2 m, parallel spacing of 0 m, wire height of 28 m and horizontal wire arrangement is horizontal. The pipeline cathodic protection potential distribution is simulated when the voltage level of the transmission line is 1000 kV, 750 kV and 500 kV respectively.

When the voltage level is different, the potential distribution pattern is approximately the same, and the potential size is different. When the voltage level is 1000 kV, the maximum potential of the pipeline is about 199 V; when the voltage level is 750 kV, the maximum potential of the pipeline is about 174 V; and when the voltage level is 500 kV, the maximum potential of the pipeline is about 116 V.

Analysis of the pipeline cathodic protection potential distribution diagrams (Figures 8–10) and their comparison diagrams (Figure 11) show that, with the remaining calculated parameters unchanged, the maximum value of the induced voltage on the pipeline appears at both ends of the pipeline, and the AC induced voltage at the middle of the pipeline is 0.

The induced voltage increases continuously from the middle of the pipeline to both ends of the pipeline. As the voltage level of the transmission line increases, the the induced voltage on the pipeline becomes stronger, and the cathodic protection potential shift becomes larger. The potential has deviated from the effective protection zone, and the cathodic protection of the pipeline fails.

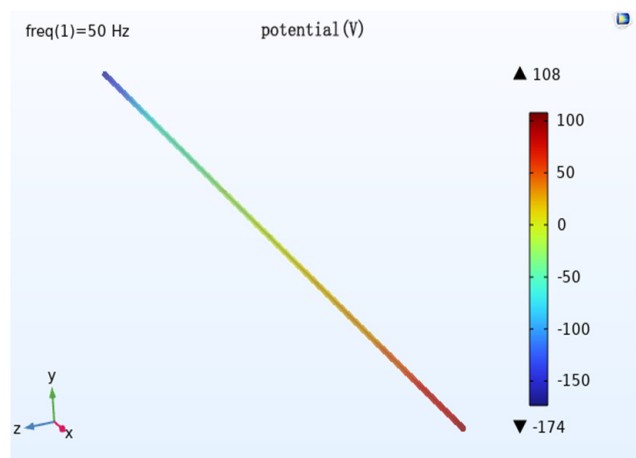

**Figure 8.** Distribution of cathodic protection potential at voltage level 1000 kV.

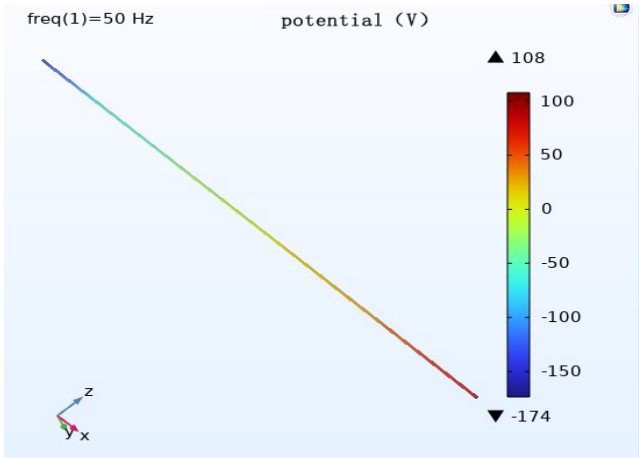

**Figure 9.** Distribution of cathodic protection potential at voltage level 750 kV.

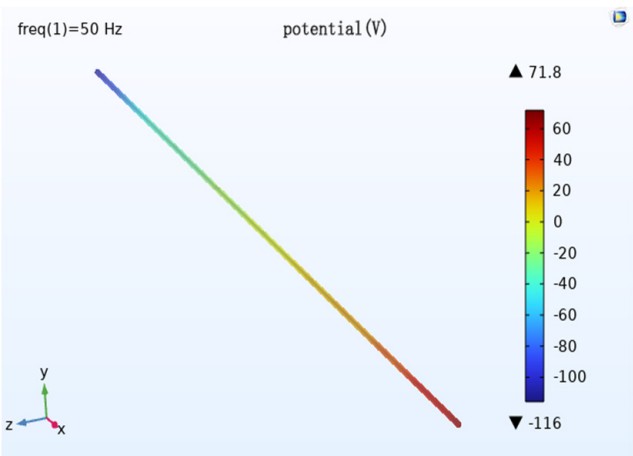

**Figure 10.** Distribution of cathodic protection potential at voltage level 500 kV.

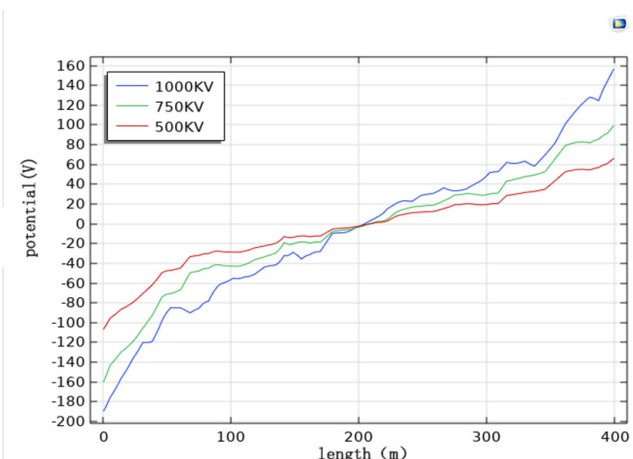

**Figure 11.** The influence law of transmission line voltage level on cathodic protection potential.

*6.3. Effect of Parallel Spacing on the Cathodic Protection Potential of Pipelines*

The parallel spacing is the distance between the transmission conductor at the vertical projection to the ground and the pipeline at the vertical projection from the ground (Figure 12).

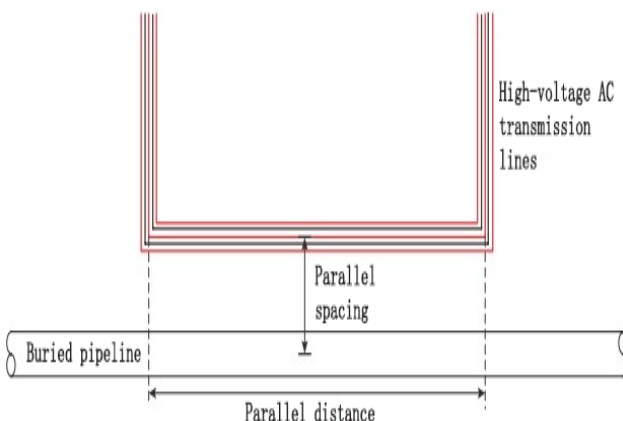

**Figure 12.** Schematic diagram of a common corridor with pipelines and transmission lines in parallel.

To study the effect of parallel spacing on cathodic protection potential, the burial depth is 2 m, the voltage level of transmission line is 1000 kV, the height of the conductor is 28 m, and the conductor arrangement is horizontal, as in the initial condition. The pipeline cathodic protection potential distribution is simulated when the parallel distance is 0 m, 10 m and 25 m, respectively.

When the parallel spacing is different, the potential distribution pattern is about the same, and the potential size is different. When the parallel spacing is 0 m, the maximum potential of the pipe is about 199 V; when the parallel spacing is 10 m, the maximum potential of the pipe is about 174 V; when the parallel spacing is 25 m, the maximum potential of the pipe is about 148 V.

Analysis of the pipeline cathodic protection potential distribution diagrams (Figures 13–15) and their comparison diagrams (Figure 16) shows that, with the remaining calculated parameters unchanged, the maximum value of the induced voltage on the pipeline appears at the two ends of the pipeline, and the AC induced voltage at the middle of the pipeline is 0. The induced voltage increases continuously from the middle of the pipeline to the two ends of the pipeline. The larger the parallel spacing, the weaker the induced voltage on the pipeline, and the smaller the cathodic protection potential offset.

The potential has deviated from the effective protection zone, and the cathodic protection of the pipeline fails.

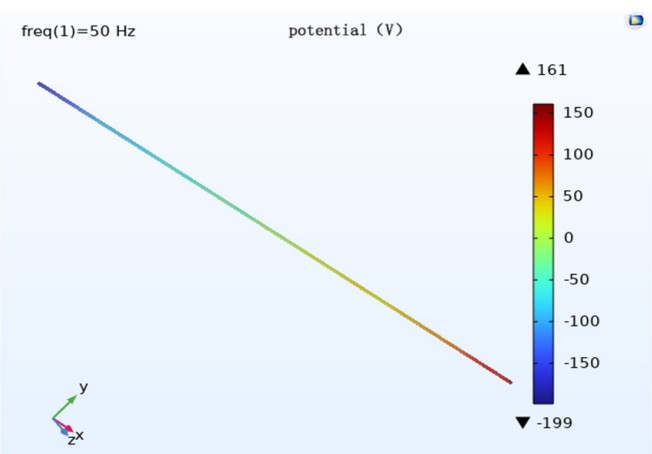

**Figure 13.** Distribution of cathodic protection potential when the parallel spacing is 0 m.

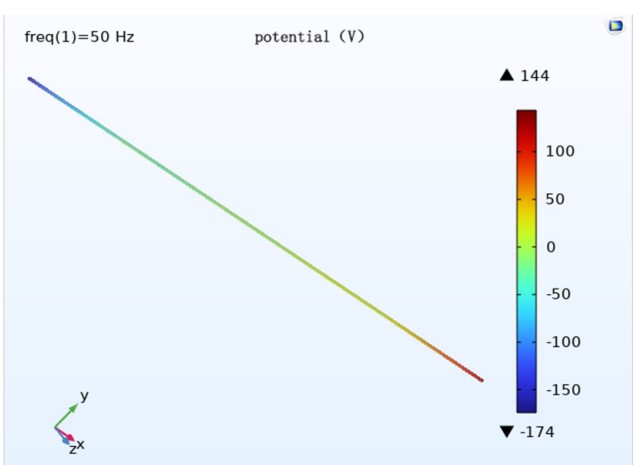

**Figure 14.** Distribution of cathodic protection potential when the parallel spacing is 10 m.

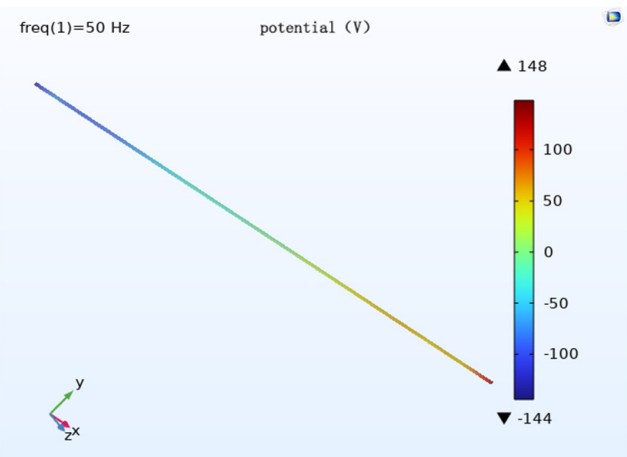

**Figure 15.** Distribution of cathodic protection potential when the parallel spacing is 25 m.

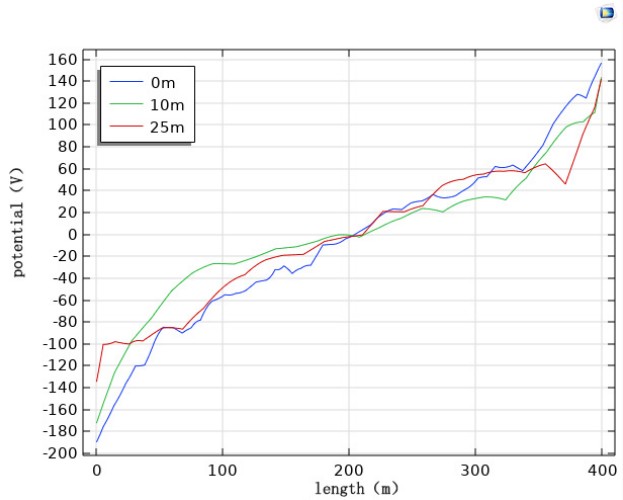

**Figure 16.** The effect law of parallel distance on the cathodic protection potential of the pipeline.

*6.4. Effect of Transmission Line Height to the Ground on Pipeline Cathodic Protection Potential*

In order to study the effect of transmission line height on cathodic protection potential, the burial depth is 2 m, the voltage level of the transmission line is 1000 kV, the parallel spacing is 0 m, and the wire arrangement is horizontal, as in the initial condition, and the pipeline cathodic protection potential distribution is simulated when the wire height is 18 m, 28 m and 38 m, respectively.

The potential size is different when the distance to the ground is different but the potential distribution law is roughly the same. When the transmission line height is 18 m, the maximum potential of the pipeline is about 230 V; when the transmission line height is 28 m, the maximum potential of the pipeline is about 199 V; when the transmission line height is 38 m, the maximum potential of the pipeline is about 160 V.

Analysis of the pipeline cathodic protection potential distribution diagram (Figures 17–19) and its comparison diagram (Figure 20) shows that the maximum value of the induced voltage on the pipeline appears at the two ends of the pipeline with the remaining calculated parameters unchanged, the AC induced voltage at the middle of the pipeline is 0, and the induced voltage increases continuously from the middle of the pipeline to the two ends of the pipeline. The greater the distance of the transmission line from the ground, the weaker the induced voltage on the pipeline, and the smaller the cathodic protection potential offset. The potential has deviated from the effective protection zone, and the pipeline cathodic protection fails.

*6.5. Influence of Wire Arrangement on Pipeline Cathodic Protection Potential*

In order to study the influence of wire arrangement on the cathodic protection potential, the initial condition is a burial depth of 2 m, transmission line's voltage level of 1000 kV, parallel spacing of 0 m, and line height of 28 m. The wire arrangement is simulated as horizontal, umbrella and triangle respectively when the pipeline cathodic protection potential is distributed.

When different wire arrangements are used, the potential distribution pattern is approximately the same, and the potential size is different. When the wire is arranged horizontally, the maximum potential of the pipe is about 234 V; when the wire is arranged triangularly, the maximum potential of the pipe is about 199 V; when the wire is arranged in an umbrella pattern, the maximum potential of the pipe is about 198 V.

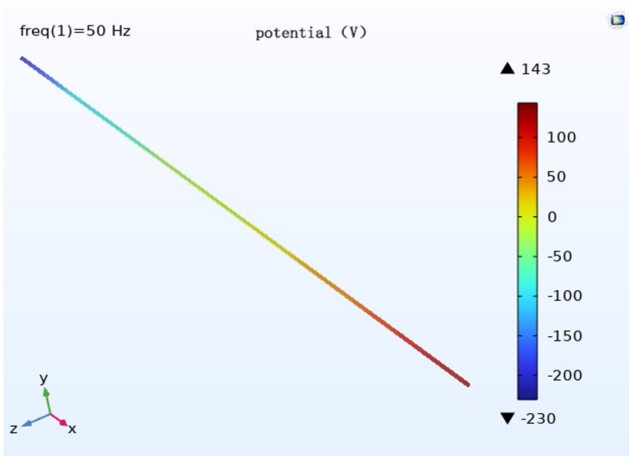

**Figure 17.** Transmission line to a ground height of 18 m when the pipeline cathodic protection potential is distributed.

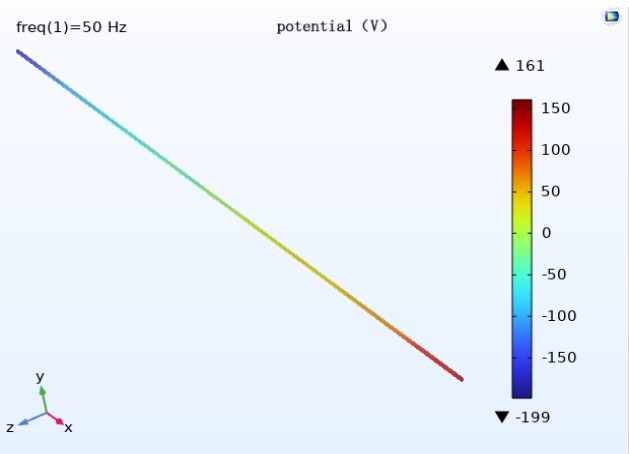

**Figure 18.** Transmission line to a ground height of 28 m when the pipeline cathodic protection potential is distributed.

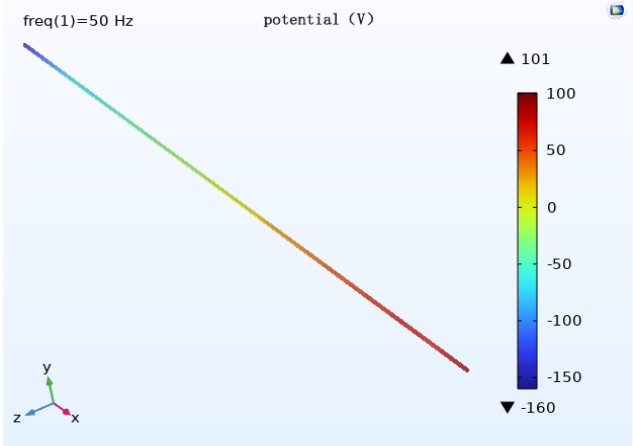

**Figure 19.** Transmission line to ground height of 38 m when the pipeline cathodic protection potential is distributed.

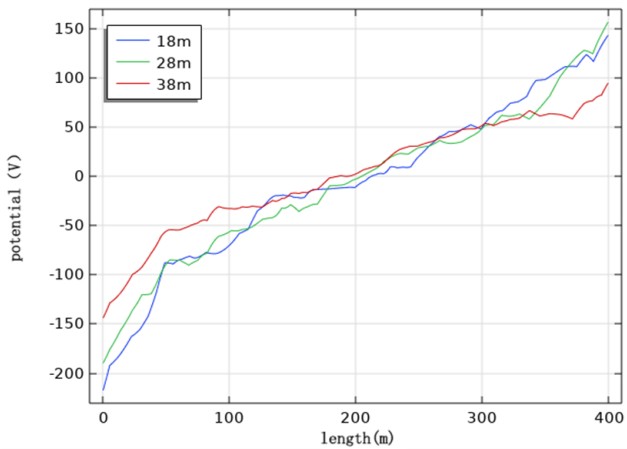

**Figure 20.** The influence of the law of transmission line height on cathodic protection potential.

Analysis of the pipeline cathodic protection potential distribution diagrams (Figures 21–23) and their comparison diagrams (Figure 24) shows that, with the remaining calculated parameters unchanged, the maximum value of the induced voltage of the pipeline appears at the two ends of the pipeline, and the AC induced voltage at the middle of the pipeline is 0. The induced voltage increases continuously from the middle of the pipeline to the two ends of the pipeline. When the wires are arranged horizontally, the pipeline cathodic protection potential deviation is the largest, followed by the triangular arrangement, and the umbrella arrangement has the least effect. The potential has deviated from the effective protection zone, and the cathodic protection of the pipeline fails.

*6.6. Effect of Burial Depth on the Cathodic Protection Potential of Pipelines*

To study the effect of burial depth on cathodic protection potential, the initial condition was a transmission line voltage level of 1000 kV, parallel spacing at 0 m, transmission line height from the ground of 28 m and horizontal wire arrangement. Simulations of the burial depth of 2 m, 4 m and 6 m were conducted when the pipeline cathodic protection potential was distributed.

When the burial depth is different, the potential distribution pattern is roughly the same, and the potential size is different. When the burial depth is 2 m, the maximum potential of the pipeline is about 199 V; when the burial depth is 4 m, the maximum potential of the pipeline is about 187 V; when the burial depth is 6 m, the maximum potential of the pipeline is about 171 V.

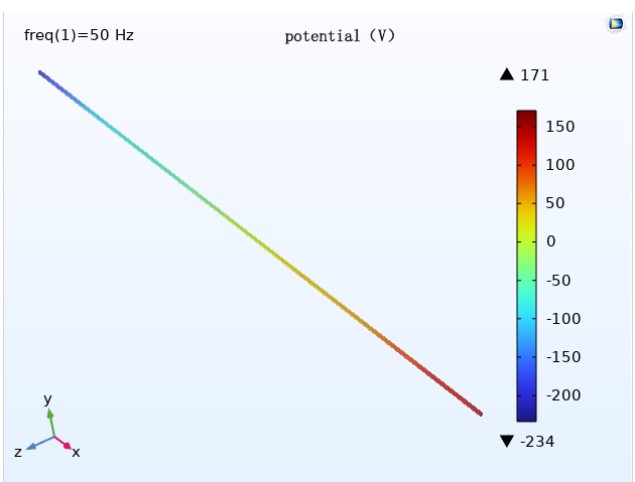

**Figure 21.** Distribution of cathodic protection potential when the conductors are arranged horizontally.

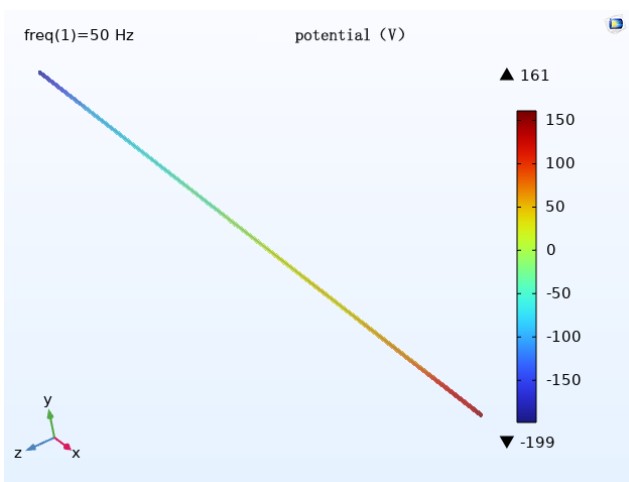

**Figure 22.** Distribution of cathodic protection potential when the conductors are triangulated.

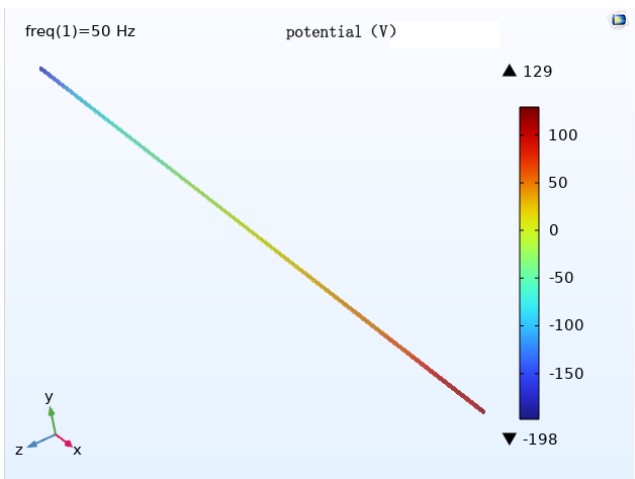

**Figure 23.** Distribution of cathodic protection potential during the umbrella arrangement of conductors.

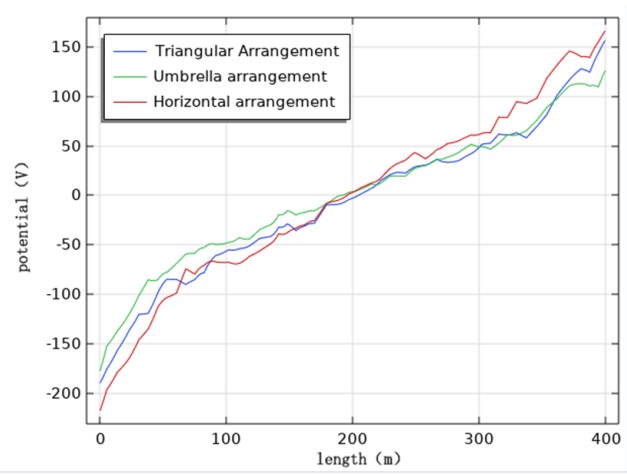

**Figure 24.** The influence law of wire arrangement on the cathodic protection potential.

Analysis of the pipeline cathodic protection potential distribution diagrams (Figures 25–27) and the comparison diagram (Figure 28) shows that the maximum value of the induced voltage on the pipeline appears at the two ends of the pipeline with the remaining calculated parameters unchanged, and the AC induced voltage at the middle

of the pipeline is 0. The induced voltage increases continuously from the middle of the pipeline to the two ends of the pipeline. The deeper the burial depth, the weaker the induced voltage on the pipeline, and the smaller the cathodic protection potential offset. The potential has deviated from the effective protection zone, and the pipeline cathodic protection fails.

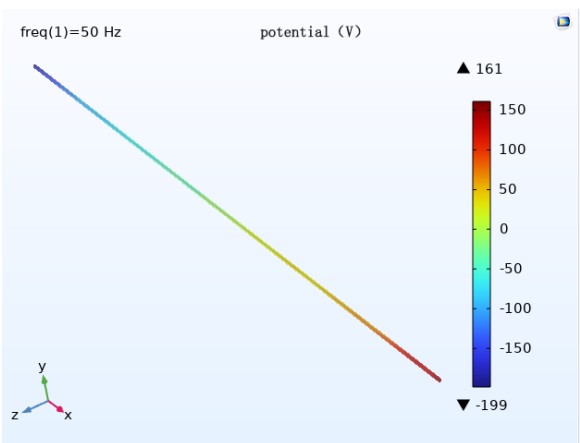

**Figure 25.** Distribution of cathodic protection potential at 2 m burial depth.

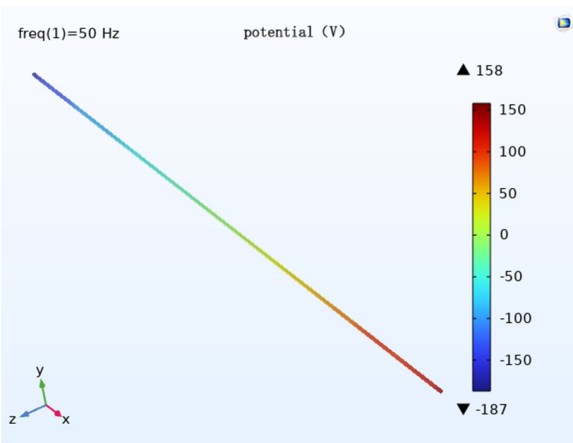

**Figure 26.** Distribution of cathodic protection potential at 4 m burial depth.

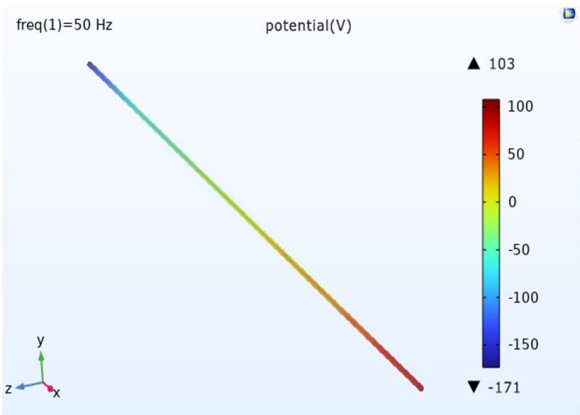

**Figure 27.** Distribution of cathodic protection potential at 6 m burial depth.

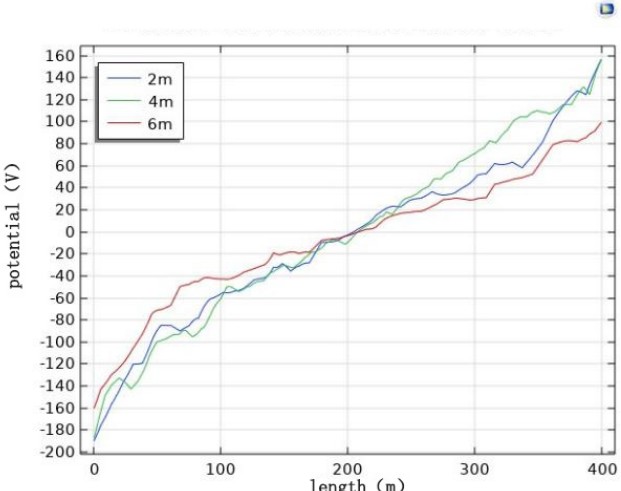

**Figure 28.** The influence law of burial depth on the cathodic protection potential.

## 7. Conclusions

Analyzing the simulation results of the effect of AC transmission lines on the cathodic protection potential of the pipeline offers several findings.

AC transmission lines have serious electromagnetic interference with the cathodic protection system of the pipeline, which can cause the cathodic protection potential of the pipeline to shift out of the effective protection zone, resulting in the failure of the cathodic protection of the pipeline.

The maximum value of the induced voltage of the pipe will appear at the two ends of the pipe, and the AC-induced voltage of the pipe in the middle position is 0. The induced voltage increases continuously from the middle position of the pipe to the two ends of the pipe, with small fluctuations in the pipe closer to the auxiliary anode, but the overall increasing trend remains the same.

The pipeline cathodic protection potential shift increases with the increase of voltage level and decreases with the increase of parallel spacing, wire height and burial depth. The pipeline cathodic protection potential shift is the largest when the wires are arranged horizontally, and the potential shift is the smallest when they are arranged in an umbrella shape.

This paper uses the COMSOL Multiphysics simulation software based on the finite element method to carry out research on the problem of electromagnetic interference from AC transmission lines to buried pipelines and accurately establishes a simulation model of electromagnetic interference from AC transmission lines to buried pipelines through the collection of actual operational data on-site. These results can be used as a basis for the design of buried lines and cathodic protection schemes for long-distance transmission pipelines.

**Author Contributions:** Conceptualization and investigation, B.Z. and Y.Z. analyzed the results and wrote the first draft of the manuscript; writing—review, B.Z. and L.L. performed the simulation calculations; supervision, resources, project administration and funding acquisition, J.W. and L.L. All authors have read and agreed to the published version of the manuscript.

**Funding:** This work was fulfilled under financial support of Xianyang Key R&D Program 2021ZDYF-GY-0023. The authors express their sincere thanks for this support.

**Institutional Review Board Statement:** Not applicable.

**Informed Consent Statement:** Not applicable.

**Data Availability Statement:** Not applicable.

**Conflicts of Interest:** The authors declare no conflict of interest.

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
