# Peer review of "Study on the Interference Law of AC Transmission Lines on the Cathodic Protection Potential of Long-Distance Transmission Pipelines"

_magnetochemistry, doi:10.3390/magnetochemistry9030075_

Round 1
Reviewer 1 Report
The manuscript entitled “Study on The Interference Law of AC Transmission Lines on The Cathodic Protection Potential of Long-distance Transmission Pipelines” analysis the impact of the overhead transmission lines on the buried pipelines. Although the topic analyzed in this manuscript is very interesting, the manuscript has a large number of shortcomings, which need to be corrected before it is considered for publication.
First of all, the abstract should be completely rewritten.
In the introductory section (more specifically in the second paragraph of the introduction) an explanation of how the capacitive coupling mechanism between transmission lines and pipelines can occur is missing.
In section 2 (applied current method), the description of the cathodic protection system does not agree with the Figure. Namely, the description mentions zinc anodes which are used in cathodic protection systems with sacrificial anodes, while Figure 2 is given for the description of impressed current cathodic protection systems. In most practical cases, in impressed current cathodic protection systems, FeSi anodes are used to protect underground pipelines. Furthermore, I feel that the description of the cathodic protection system is completely superficial in this section.
In the description of the used method, explanation on how infinite boundaries were taken into account should be given, since the soil and air are semi-infinite domains. From the Figure 4 it can be noted that the computational domain (soil and air) is limited.
Tables 1 and 2 are not properly formatted.
In the manuscript, it is necessary to give and describe all used boundary conditions. Also, it is of crucial importance to give the parameters that describe the electrode kinetics, as well as the formulas that describe the electrode kinetics. A brief description of how this (electrode kinetics) is implemented in COMSOL would be desirable.
It is necessary to clearly indicate the novelty of this manuscript in the introductory and concluding sections.
Reviewer 2 Report
The article presents the study of the influence of AC transmission line onto the cathodic protection of the buried pipelines. In general the problem is impotant and interesting, however I find this manuscript now as in a very low quality.
1. From the Introduction, the reader cannot find what is the novelty of your method. What is the difference between FEM and the rest of the methods mentioned in the Introduction? What are the advantages and disadvantages?
2. Considering the FEM model as the novelty, there must be its validation. The validation of the FEM model can be prepared against measurements (what is difficult in this case) or against other theoretical method(s) (e.g. those mentioned in the introduction). Without the validation (comparison with other well known methods), nobody knows if this model is properly set and if the results are reliable.
3. The FEM model is described nowhere. There is no information about used modules (and their equations), used solvers. I do not know the frequency, boundary conditions etc. The number of mesh elements equal to about only 56 000 what is extremely scarse for such a big domains. It must be described in details.
4. There are many mistakes from the electrical and electromagnetic point of view. For example potential vs. voltage. I do not know which potential is which in this simulation. In my opinion, one part of the system must be grounded (set as potential equal to 0) and from now it is understandable what the rest of "potentials" means. Otherwise, the potential has no meaning here. The second thing is that more important is the current going through the high voltage grid onto the magnetic coupling because the current is responsible for the magnetic field. Voltage is responsible only for the electric field what has influence onto the capacitive coupling. Another: The "voltage level" is different in each of the 3-phase overhead lines at the time due to the phase shift between them. So the simulation should take it into account. end so on...
Concluding, the manuscript has a potential, however the authors must provide profound major revision.
Some detailed comments:
Line 14: typoo error
Line 49: space after dot
Line 51: space after dot - please check the whole article
Line 63: interference?
Line 63: space before "(" and after ")"
Line 64: space after ";" - please check the whole article
Line 96: volts "V" must be written by the capital letter
The beginning of the chapter 3 looks like an advertisement. I think the information that it is FEM software and the reference to the COMSOL webpage is enough.
Please use space between number and the unit.
Lines 119-121: what is the base of assuming of those dimentions? I do not understand those sizes (what is the length, width, radius etc.). The dimentions of the ambient must be chosen depending on the frequency of the electromagnetic field. It has significant impact onto the results.
Chapter 3.2: I do not understand what does mean "for refinement". Do you mean do you use adaptive meshing? Please provide information about adaptive meshing technique e.g. from [doi.org/10.1016/j.ijepes.2021.107737].
Table 1: I do not understand this table. What does mean "load current". Why do you use slash before unit? There should be comma. What does mean "transmission voltage level"?
Table 2: it is illegible. This table should be widen to fit the captions of the columns.
Chapter 4: The solver used for the simulation is described nowhere. Is it transient? Steady state? What kind of simulations AC/DC module? What is the frequency? etc.
Begining of the chapter 4: I do not understand this voltage level. Is it a voltage level of the grid over the soil? Is is a DC voltage or AC? if AC, is it 50 or 60 Hz?
The pipeline should be closed in this simulation, otherwise it is an open circuit.
Fig. 9: What voltage is this? Is it induced voltage?
Similar comments to the rest of the article.
Round 2
Reviewer 1 Report
The authors have partially corrected the first version of the paper, but there are still unclear and unfinished things to be done before the manuscript is ready for publication. First of all, although I suggested, the authors of the manuscript did not at the end of the introductory section provide an manuscript overview.
Section 2 heading is missing.
Keywords are missing.
There are still a lot of typographical errors in the current manuscript.
It is still not resolved what and how the boundary conditions were used. Which boundary condition was used on the surface of the high-voltage overhead line, which on the soil/air boundary surface, which on the anode surface and which on the cathode surface.
Although the parameters are given in Table 4, it is not clearly visible what type of boundary conditions were used on the electrode surfaces. The boundary conditions (polarizing curves) on the electrode surfaces of the cathodic protection system can have different mathematical forms.
My opinion is that the connection between the mathematical models given in section 4.1 and 6.1.1 is not clearly visible.
Reviewer 2 Report
The article still needs many improvements. The most important are:
1. The soil has its own conductivity whose value is not given in the manuscript. This value depends on the moisture and soil composition, which significantly reduce the voltage induced along the gas pipe. It must be given with reference(s).
2. Lines 286-288: "The geometric model is meshed in COMSOL using a physical field control mesh with a hyper-detailed cell size and a complete mesh containing "679193" domain cells, 36805... - some of the numbers have " " while some do not. I still do not agree with the number of mesh elements. You should use adaptive meshing tool to adjust properly the number and the location of mesh elements. Otherwise, how do you know if it is proper and how do you know the error value?
3. Although authors gave information in the reply letter about incorporation of the phase shift of the HV lines to the manuscript, I do not see it in the new version of the paper.
4. Authors provided many equations whose source (references) is unknown. Moreover, the letters in most of the equations are described nowhere.
5. There is still a problem with definition of the 0 (zero, ground) potential. As I told the authors last time, one of the boundary must be set as ground with zero potential. Otherwise, the voiltage potentials looks random (e.g. what 0 potential means now?).
6. All of the equations are formatted wrongly. Why there is so space between letters in equations?
7. The article still contains many typoos and formating issues e.g. lack of spaces befor [], before dots, some of the captions are in not the same page as the figure e.g. Fig. 4. Some of the figures are illegible (e.g. Fig. 5). Authors use different formating for units (e.g. sometimes mA/m while sometimes E-3 A/m etc.)
8. I still do not see the novelty in this manuscript.
Round 3
Reviewer 1 Report
The authors adequately responded to my previous objections. My opinion is that the quality of manuscript has been improved. It can be noticed that the formulas font is not adequate. Also, my opinion is that the title 6.1.3 is completely redundant and that the table from this section naturally continues to the text of section 6.1.2, and also there is no need to create a section that contains only a table. All this can be corrected in the editing stage.
Reviewer 2 Report
After second revision, I still do not see the novelty of this manuscript. My main concerns are:
1. The authors claim that the novelty of their work is based upon preparing of the FEM model of the induced voltage along the buried pipes, while the other currently published works recommend MoM. The authors do not explain in the manuscript, what are the advantages of their FEM model over MoM. The presented FEM model requires commercial (expensive) FEM software, while MoM can be easily implemented into e.g. free Octave or C++ software. Similarly, MoM is typically computed much faster than FEM. So why the authors prepared their model?
2. Even if we assume that only FEM model is the novelty (which is very weak - the model is very simple), we do not see the advantages over another algorithms. There should be given comparison between time of computation, hardware requirements and the accuracy of the results.
3. The results obtained by FEM are neithter validated by any measurements nor other currently existing models. We do not know now, if this model is even properly set.
